# Lung Adenocarcinoma Tumor Origin: A Guide for Personalized Medicine

**DOI:** 10.3390/cancers14071759

**Published:** 2022-03-30

**Authors:** Laetitia Seguin, Manon Durandy, Chloe C. Feral

**Affiliations:** Université Côte d’Azur, INSERM, CNRS, IRCAN, FHU Oncoage, 06107 Nice, France; manon.durandy@etu.univ-cotedazur.fr

**Keywords:** LUAD, cell of origin, lung progenitor, oncogenic driver, immune infiltration

## Abstract

**Simple Summary:**

Lung cancer is the leading cause of cancer-related death worldwide, with an average 5-year survival rate of approximately 15%. Among the multiple histological type of lung cancer, adenocarcinoma is the most common. Adenocarcinoma is characterized by a high degree of heterogeneity at many levels, including histological, cellular, and molecular. Understanding the cell of origin of adenocarcinoma, and the molecular changes during tumor progression, will allow better therapeutic strategies.

**Abstract:**

Lung adenocarcinoma, the major form of lung cancer, is the deadliest cancer worldwide, due to its late diagnosis and its high heterogeneity. Indeed, lung adenocarcinoma exhibits pronounced inter- and intra-tumor heterogeneity cofounding precision medicine. Tumor heterogeneity is a clinical challenge driving tumor progression and drug resistance. Several key pieces of evidence demonstrated that lung adenocarcinoma results from the transformation of progenitor cells that accumulate genetic abnormalities. Thus, a better understanding of the cell of origin of lung adenocarcinoma represents an opportunity to unveil new therapeutic alternatives and stratify patient tumors. While the lung is remarkably quiescent during homeostasis, it presents an extensive ability to respond to injury and regenerate lost or damaged cells. As the lung is constantly exposed to potential insult, its regenerative potential is assured by several stem and progenitor cells. These can be induced to proliferate in response to injury as well as differentiate into multiple cell types. A better understanding of how genetic alterations and perturbed microenvironments impact progenitor-mediated tumorigenesis and treatment response is of the utmost importance to develop new therapeutic opportunities.

## 1. Introduction

Lung cancer is the third most common cancer and the leading cause of cancer death worldwide. It is a highly heterogeneous disease historically classified based on tumor histology into two main types: small cell lung carcinoma (SCLC), which encompasses 15% of all lung cancers, and non-small cell lung carcinoma (NSCLC). NSCLC subdivided into several histological subtypes including adenocarcinoma (LUAD), adenosquamous carcinoma, squamous cell carcinoma (LUSC) and large cell carcinoma (LCC) [1]. LUAD is by far the most common subtype of NSCLC (50%) and arises in more distal airways [2,3]. Although the most common cause of this cancer is chronic exposure to tobacco smoke, its incidence in non-smokers accounts for from 15–20% of cases and is often attributed to a combination of genetic and environmental factors [1]. Despite the emergence of promising new therapies, such as immunotherapy and targeted therapies, LUAD remains a major public health problem with a 5-year survival rate of 15% due to the late stage of diagnosis and the innate or acquired resistance of these tumors to anti-cancer treatments [1]. One of the difficulties in the treatment of LUAD is the great intra- and inter-tumor heterogeneity highlighting the need to determine the different cellular and molecular factors of LUAD.

Tumor heterogeneity could be attributed to genetic and non-genetic mechanisms [4]. Genomic instability, including somatic mutations, amplifications, copy number gains and chromosome rearrangements, represents one of the hallmarks of cancer and results in genetic aberrations. Integrated analysis of the genome and transcriptome has revealed that human LUAD tumors are characterized by a high tumor mutational burden (TMB) compared to other cancer types, probably related to carcinogen exposure [5]. Strong evidence supports the notion that cell of origin patterns dominate the molecular classification of cancers [6]. In particular, human LUAD tumors arise from progenitor clones bearing early genomic alterations that drive tumorigenesis [6,7,8,9]. Resulting subclonal populations will evolve by adapting to microenvironmental pressures leading to alterations in cell identity and lineage control [7].

Large-scale genomic studies of human tumor biopsies have involved several genetic alterations in the initiation of LUAD such as *TP53*, *KRAS*, *KEAP1*, *STK11*, *ALK* and *EGFR* [10,11,12]. One idea of precision therapy is to identify subgroups of cancer patients that will benefit from specific therapeutic strategies. Therefore, finding druggable target molecules to inhibit oncogenic signaling is a key challenge in cancer research. Advances in drug development and pharmacogenomics combined with the molecular characterization of tumors have allowed the emergence of promising targeted therapies for EGFR- and ALK-driven cancer patients [1]. Mutations in the *KRAS* oncogene represent one of the most prevalent genetic alterations in LUAD and patients whose tumors harbor *KRAS* mutations typically have a poor prognosis, with very aggressive tumors [13,14]. Despite multiple approaches developed so far to interfere with mutant KRAS and its downstream signaling, and recent progress in specifically targeting KRAS^G12C^ mutant, most strategies have failed in clinical settings [15]. Importantly, in pancreatic and lung mouse cancer models, introducing *Kras* and *Tp53* mutations in epithelial cells produce distinct outcomes [8,16,17], suggesting that microenvironmental factors may influence a cell’s response to an oncogenic stimulus. Thus, defining the cell of origin of LUAD can uncover the mechanisms of tumor initiation and progression and identify unique type-specific targets for therapy [18,19].

## 2. Defining Lung Progenitor Cells

The lung is a highly quiescent tissue with a regenerative turnover approximately seven years but has a remarkable reparative capacity under injury. As the primary function of the lung is to facilitate air/blood gas exchange, it is frequently exposed to potential epithelium-damaging stimuli, including pathogens, cigarette smoke and airborne pollutants, that challenge it to rapidly repair itself [20,21]. Multiple stem or progenitor cells with multipotency are required for maintaining lung functions during normal conditions and repair depending on the type and severity of the injury [20]. Developmental processes underlying normal tissue regeneration have been implicated in cancer. Thus, understanding lung tissue repair will underline carcinogenesis complexity.

The adult lung is composed of branching airway networks, including from the proximal to distal: the trachea, bronchi, bronchioles, and alveoli. Distally, the airways open into the alveoli at the bronchioalveolar duct junctions (BADJs) in the mouse, or into respiratory bronchioles, which contain multiple alveolar ducts in humans [20,22]. The alveolar space is composed of alveolar type 2 (AT2) cells, specialized for pulmonary surfactant production to reduce alveolar surface tension during respiration and alveolar type 1 (AT1) cells in close apposition to capillaries, specialized for gas exchange [22] (Figure 1).

Several region-specific epithelial stem/progenitor cells have been identified in mice, including basal cells in the proximal airways, neuroendocrine cells and variant Club (vClub) cells in the bronchioles, bronchioalveolar stem cells (BASCs) in the BADJ and AT2 cells in the alveolar space [20] (Figure 2). Extensive works using multiple injury models targeting specific lung regions have shown that basal cell, vClub cells (former Clara cells expressing Clara cell 10-kd protein CC10) and AT2 cells are progenitors for the lung trachea, bronchi and alveoli, respectively [20,22].

In particular, AT2 cells characterized by their surfactant protein C (SPC) expression are able to self-renew and to slowly differentiate into AT1 cells over approximately a year in homeostatic conditions but undergo rapid clonal expansion and differentiation to AT1 cells after specific ablation of in response to bleomycin-induced alveolar injury in mice [23]. However, lineage-tracing of SPC^+^ AT2 revealed that most of the newly generated AT2 cells were derived from SPC^−^ cells under bleomycin injury, suggesting that other progenitors are involved in distal lung repair [23].

With the emergence of single-cell RNA sequencing technology, a complex picture of alveoli regeneration has been proposed with the discovery of a Wnt-responsive alveolar epithelial progenitor (AEP) lineage within the AT2 cell population that acts as a major facultative progenitor cell in the distal lung under acute influenza-induced mouse lung injury. The AEPs population has been validated in human lungs and represent 30% of human AT2 cells. They are characterized by the expression of Axin-2 and the conserved cell surface marker transmembrane 4 superfamily, also known as the tetraspanin family (TM4SF1) [24,25]. AEPs are able to self-renew to maintain the AEP lineage and generate a large number of a new lineage-traced alveolar epithelial progeny in response to influenza-induced mouse lung injury.

Interestingly, mature AT2 are also able to acquire the AEP fate, suggesting a lung plasticity under acute alveolar injury [24,25]. The transdifferentiation of cuboid AT2 into thin AT1 require extensive stretching, making them vulnerable to DNA damage. Two studies revealed that AT2 undergoing differentiation toward AT1 adopt pre-alveolar AT1 transitional cell states [26,27]. While these cells are extremely rare in steady state, they are significantly induced after injury. These damage-associated transient progenitors called DATPs [26] or PATs [27] are distinct from AT2 and show an enrichment of TP53, TGFβ, DNA-damage-response signaling and cellular senescence. AT2-to-AT1 transdifferentiation is dictated by TP53 activation and inflammatory stimuli within the alveolar space. Indeed, interstitial macrophage-derived IL-1β controls DATP transition but prevents AT1 differentiation, leading to the aberrant accumulation of DATPs and impaired alveolar regeneration in mouse [26]. The accumulation of these transitional state cells in human fibrotic lungs suggests persistence of the transitional state in disease conditions [27].

In addition, AT2-to-AT1 trajectory modeling with longitudinal single-cell RNAseq together with lineage-tracing during epithelium repair, revealed that airway and alveolar stem cells converge on a unique Krt8^+^ transitional stem cell state during murine alveolar regeneration [28]. These cells derived from activated AT2 are able to give rise to AT1. They have squamous morphology, display features of cellular senescence and exhibit a distinct cell–cell communication network with mesenchyme and macrophages during tissue regeneration [28]. The Krt8^+^ transitional stem cell state does not resemble a linear gene expression intermediate from stem cells towards AT1. It pre-exists in normal conditions suggesting that these cells are derived from AT2, possibly during normal homeostatic turnover [28]. In addition, human AT2, unlike murine AT2, are able to transdifferentiate into metaplastic Krt5^+^ basal cells under inflammatory stimulation [29]. This process involves an alveolar-basal intermediate state resembling the Krt8^+^ transitional stem cell state, suggesting that a Krt8^+^ intermediate state may not be unique during AT2-to-AT1 transdifferentiation in mice, but also evolved to include AT2-to-basal transdifferentiation in humans [29].

Besides AT2, other progenitors clearly contribute to alveolar repair following murine lung injury. At the BADJ, BASCs have been identified in 2005 by Kim et al. [30] and express both the Club cell marker, CC10 and the AT2 cell marker, SPC. Under lung injury, BASCs are capable of self-renewal and multipotent differentiation into Club, AT1 and AT2 cells, thus, contributing to the regeneration of both of the distal airways [30]. Their function as a dual progenitor has remained debated until recently, with a genetic lineage-tracing experiment being performed using dual recombinases (Cre and Dre) to specifically track BASCs in vivo [31]. Fate mapping and clonal analysis showed that BASCs rarely give rise to AT1 and AT2 cells during lung homeostasis but are activated and respond distinctly to different lung injuries by transdifferentiating into multiple lineages [31,32].

BASCs fate determination is dictated by the microenvironmental niche. Indeed, Thrombospondin-1 (TSP1), an angiogenesis inhibitor, highly expressed in lung endothelial cells [33], directs BASCs differentiation toward the alveolar lineage [34]. While BASCs were first isolated based on the expression of surface markers EpCAM, CD34 and Sca-1, further refinement of their surface markers suggested that BASCs were enriched in the EpCAM^hi^ integrin α6β4^+^ (CD49f/CD104) Sca-1^lo^CD24^lo^ subset questioning their cell surface marker identity [35,36]. Transcriptional analysis of BASCs not only confirmed that their gene profiles partly overlapped with AT2 and club cell gene profiles but also demonstrated that they can be clustered into two distinct subpopulations [31]. This heterogeneity may explain, at least partially, controversial results.

Rare club-like lineage-negative epithelial stem/progenitor (LNEPs) cells, present within a normal distal lung, are also able to regenerate an injured murine lung. These cells also express integrin α6β4 (CD49f/CD104) but no well-defined lineage markers such as SPC and CC10. In vivo, cells expressing integrin β4^+^/SPC^−^ are able to expand clonally and to differentiate toward mature CC10^+^ airway-like and SPC^+^ saccular structures when implanted in the kidney capsule [37]. In addition, they are able to proliferate and repopulate lung alveolar space after bleomycin-induced lung injury [37]. Further studies demonstrated that these rare LNEPs are quiescent during homeostasis mobilized to regenerate murine lung epithelium after major injury such as influenza [38,39].

LNEPs are subdivided into two distinct LNEP subtypes: the p63^+^ LNEPs and the p63^−^ LNEPs. p63^+^ LNEPs are defined by the high expression of CD14 and activate the ΔNp63/Krt5^+^ remodeling program after bleomycin or influenza injury. They proliferate and migrate to occupy injured areas and differentiate toward mature epithelium [40].

Mechanistically, single-cell RNA-Seq analysis of primary human lung epithelial cells from normal and fibrotic lungs indicates that hypoxia/Notch signaling promotes AEC2s transdifferentiation towards basal-like cells after major injury [40]. Indeed, local lung hypoxia through hypoxia-inducible factor (HIF1α), drives Notch signaling, which activates the ΔNp63/Krt5^+^ program and basal-like cell expansion, whereas the subsequent Notch blockade promotes rapid cell differentiation into AT2 and migration within the alveolar space [38,40]. Krt5^+^ basal cells migrate to damaged alveolar regions and transdifferentiate into AT2 and AT1 to recreate an epithelial barrier under influenza-induced acute lung injury [38,40,41,42,43]. The p63^−^ LNEPs appear to be the major airway subpopulation expressing the cell surface marker Sca-1. This subset represents 5% of LNEPs and is distinct from BASCs by their bronchiolar localization. Under bleomycin-induced lung injury β4^+^/p63^−^ LNEPs support AT2 and AT1 cell differentiation and alveolar regeneration [39]. These cells express high levels of major histocompatibility complex (MHC) class I and II genes, including *H2-K1*, involved in the immune response to infections. Interestingly, they also show a high level of genes supporting viral replication and are preferentially targeted during influenza injury, suggesting that the nature of injury determine the progenitor response.

Overall, these findings support the paradigm that multiple pre-existing specialized stem/progenitor cells preferentially mobilize following injury to repopulate the lung epithelium. The particular type of responder is strongly influenced by the nature and the intensity of the injury and required a dialogue with the surrounding microenvironment, including developmental and inflammatory factors. Considering the vital importance of rapid and proper tissue repair, cell plasticity and progenitor redundancy appear as a survival mechanism to induce efficient lung regeneration. Moreover, the distinctive biology of pre-existing different lung stem cells could drive the distinct phenotypes and genotypes of tumors, resulting in heterogeneity since the tumor initiation.

## 3. Cell of Origin of LUAD: Multiple Possibilities

Tumors have long been suspected of hijacking stem cell mechanisms used for tissue maintenance and repair. Lung cancer is generally thought to originate from the malignant transformation of adult lung stem cells [44,45], but cell of origin of LUAD is still under debate. Most of the knowledge about the LUAD cell of origin is based on the use of genetically engineered mouse models (GEMMs) allowing for spatial and temporal control of oncogene activation and tumor suppressor inhibition [46] (Table 1). Since activating the *KRAS* mutation is a key initial event in LUAD tumorigenesis, several transgenic mouse models permitting the induction of the *Kras* mutation in the lung have been generated.

The most commonly use GEMM is the *Kras^LSLG12D^* model in which LUAD occurs after the induction of the *Kras* mutation by intratracheal or intranasal instillation of recombinant adenoviral Cre [54]. To clearly determine the LUAD cell of origin, cell type-specific Cre drivers with engineered knock-in strategies or with adenoviruses driving Cre expression by specific promoters have been extensively used [46]. However, results are elusive and LUAD precursors differ depending on which Cre delivery system has been used. While AT2 transformation has been unequivocally involved in LUAD development with tamoxifen or adenovirus delivery systems, the role of BASCs and club cells is still unclear (Table 1).

Indeed, it has been shown that BASCs can proliferate in vitro and are expanded at the early stages of tumorigenesis in vivo following *Kras^G12D^* mutation [30]. However, when the *Kras^G12D^* mutation is specifically induced by 4-OH-tamoxifen treatment in either alveolar SPC^+^ cells or bronchiolar CC10^+^ cells, specific SPC^+^-driven *Kras* mutation leads to LUAD while CC10^+^-driven *Kras* mutation leads to hyperplasia in the bronchioalveolar duct region. These data suggest that club cells and BASCs are not the cell of origin of murine LUAD [47].

This discrepancy may be explained by the experimental context and the lack of specific markers to discriminate these populations. First, adenoviral infection induces an inflammation required for club cells and BASCs to transform [30]. This hypothesis has been tested by monitoring lung cells at the single-cell level using a mouse model in which the *Kras^G12V^* mutation is induced by 4-OH-tamoxifen treatment in a very limited number of adult lung cells and visualized by X-Gal staining, a surrogate marker coexpressed with *Kras^G12V^* [48]. Only AT2 cells were able to form LUAD. However, under inflammatory stimulation with adenovirus infection, bronchiolar cells were able to form adenomas. In addition, the expression of *Kras^G12V^* under the control of Sca-1 promoter during embryonic development induces CC10^+^ tumors. These results demonstrate that BASC and AT2 respond differently to *Kras* oncogenic signals. While AT2 are able to transform regardless of their surroundings, CC10^+^ cells require additional triggers [48]. This observation is reinforced by another study combining a chemically-induced LUAD model, where mice are exposed to toxic chemicals found in tobacco smoke, with genetically engineered reporters allowing cell lineage tracing in vivo [49]. Airway epithelial cells not only gained *Kras* mutations after toxic exposure but also migrate to the alveolar space and transdifferentiate into alveolar cells, thereby replenishing this population under lung injured conditions. The removal of these airway epithelial cells before toxic exposure leads to the inhibition of LUAD formation. Altogether, these results imply that tobacco-induced LUAD may start in the airway epithelial cells [49]. Second, the lack of a specific marker to discriminate BASCs from AT2 cells and cellular plasticity during LUAD formation may explain, in part, the elusive results. Discrimination of alveolar (EPCAM^+^, MHCII^+^) cells and bronchioalveolar (EPCAM^+^, MHCII^−^) cells showed a tumorigenic potential enrichment of the bronchioalveolar cells. In addition, bronchiolar cell-derived secondary tumors developed in both the bronchiolar and alveolar regions of recipient mice, whereas alveolar cell-derived secondary tumors occurred mainly in the murine alveolar region [55]. Furthermore, CC10-driven oncogenic *Kras* expression led to CC10^+^ hyperplastic cells giving rise to SPC^+^ adenoma by gradually losing CC10 expression and acquiring SPC [50].

While *KRAS* mutation is prevalent in smokers with chronic inflammation, *EGFR* mutation occurs preferentially in non-smokers. Consistent with the hypothesis that CC10^+^ cells require an inflammatory microenvironment to transform, *Egfr* mutant-driven lung cancer leads to an almost exclusively alveolar type LUAD, with a rare tumor exhibiting bronchiolar features [56]. Bronchiolar and alveolar tumor organoids have distinct drug responses highlighting the importance of cell of origin in cancer therapy response. In addition, transcriptional analysis demonstrates that bronchiolar tumoroids have transcriptional enrichment in TNF-*α*, KRAS, mutated TP53, EGFR, BMI1 deletion and PRC2 activity associated pathways compared to alveolar tumoroids that express higher E2F targets and G2/M checkpoint genes, suggesting that oncogenic transformation drives distinct transcriptional landscapes in cells with different cellular origins [56].

Altogether, these results revealed that LUAD can arise from different progenitor cell populations depending on the microenvironmental context and oncogenic driver.

## 4. Cell of Origin of LUAD: The Influence of Co-Occurring Mutations

Tobacco smoke contains many toxic, carcinogenic and mutagenic chemicals that exert their genotoxic effect by causing double-stranded DNA breaks and the generation of reactive oxygen species (ROS), causing mutation in vital genes [9,57]. Molecular studies of the development of LUAD have identified several *KRAS* co-occurring tumor suppressor gene abnormalities prevalent in patients with smoking history, such as those in *TP53*, *STK11* and *KEAP1* [10]. The p53 protein is the most frequently mutated tumor suppressor in cancer. p53 can trigger a variety of anti-proliferative programs by activating or repressing key effector genes [58]. The conditional loss of p53 in *Kras^G12D^;Tp53^fl/fl^* mice caused accelerated tumor development with higher grade and increased metastases [59]. Interestingly, the use of progenitor cell-type-restricted adenoCre targeting alveolar or bronchiolar cells demonstrated that both progenitors distinctly contribute to LUAD tumorigenesis, evidenced by differential histopathological spectra [50]. AT2-driven tumors are aggressive and metastatic while CC10-driven tumors exhibit more pronounced papillary features [50] (Table 1).

Tumor protein kinase Ci (PKCi) is also a key oncogenic factor in human LUAD, frequently overexpressed in primary tumors [60]. While genetic ablation of *Prkci* (encoding PKCi protein) block BASCs expansion and subsequent lung adenoma formation in response to oncogenic *Kras* [60], recent work demonstrated that PKCi expression favors BASCs-derived LUAD and PKCi-dependent tumors through the activation of PKCi/ELF3-NOTCH3 signaling, while *Prkci* deletion leads to Axin2^+^ AT2-derived LUAD that are PKCi-independent and exhibit dependency upon Wnt/β-catenin signaling for their growth and stem-like activity [51]. Consistent with their different origin and distinct oncogenic signaling mechanisms, BASC-derived and Axin2^+^ AT2-derived LUADs exhibit distinct sensitivity to pharmacologic inhibition of PKCi and Wnt signaling [51]. Thus, the subset of primary human LUAD tumors with elevated and nuclear expression of β-catenin in association with low PKCi expression may exhibit sensitivity to Wnt/β-catenin-targeted therapies (Table 1).

*KRAS* mutation concurrent with the loss of *STK11* is detected in 30% of LUAD and represents a more aggressive subtype recapitulated by the preclinical *Kras^G12D^, Stk11^fl/fl^* mouse model [52,61]. *STK11* is a cancer-suppressive gene encoding LKB1 protein that acts as a critical regulator of cellular metabolism and energy sensing by activating AMP kinase (AMPK). Once activated, AMPK inhibits mTOR signaling. *Stk11* mutation (and the consequent LKB1 loss) directly induces epigenetic changes and confers a proliferative advantage to cancer cells. LKB1 inactivation leads to the development of a mixture of adenocarcinoma, adenosquamous carcinoma and squamous cell carcinomas in mice [62,63]. *Stk11* deletion under progenitor cell-type-restricted adenoviral Cre revealed that CC10^+^ cells are the predominant progenitors of adenosquamous and squamous carcinoma while SPC^+^ cells only develop typical LUAD. This suggests that the role of LKB1 is restricted to airway cells and, therefore, its loss in alveolar cells does not significantly affect LUAD originating from AT2 cells [52] (Table 1). As LKB1-deficient cells generate elevated levels of reactive oxygen species [64,65], AT2-derived adenocarcinomas tend to transdifferentiate to adenosquamous carcinoma. This suggests that microenvironmental pressures can modulate features imposed by the cell of origin [64].

In addition, concurrent loss-of-function mutation in *Keap1* drives ferroptosis protection in LKB1 deficient tumors [66]. KEAP1 is a key member of the antioxidant response pathway, functioning as a negative regulator of nuclear factor erythroid 2-related factor (NRF2) [67]. Loss of function mutations in *KEAP1* activates the NRF2 pathway that is essential for sensing oxidative stress and protecting cells against ROS. Co-mutations of *KRAS* and *KEAP1* represent from 10 to 20% of human LUAD. *Kras*-driven LUAD with loss of *Keap1* arises from bronchiolar cell of origin, likely due to an increased susceptibility of bronchiolar cells to enhanced NRF2 pathway activity [53] (Table 1). *Keap1*-mutated tumors are dependent on glutaminolysis [67] and activate the pentose phosphate pathway, the inhibition of which abrogated tumor growth [53]. These studies highlight alternative therapeutic approaches to specifically target this unique subset of *KRAS*-mutant LUAD cancers.

Altogether, these results suggest the cell-of-origin combined with the genetic alteration profile of the tumor can dictate the LUAD histopathological phenotype and specific vulnerabilities.

Overall, these findings provide evidence that lung cancer heterogeneity depends on the oncogenic driver, co-occurring mutation and the cell of origin. Thus, understanding how intrinsic factors and microenvironment pressures will lead to cellular transformation and how subsequent cancer cells will hijack signaling pathways in their cell of origin will allow opportunities to tailor therapies for subsets of LUAD.

## 5. Cell of Origin of LUAD and CSCs Relationship

It is now well-known that LUAD is maintained by a subpopulation of cells, the cancer stem cells (CSCs) that correlate with tumor aggressiveness and therapy resistance [19,68,69]. CSCs are characterized by their tumor initiation capacity and share several features of normal stem cells. They can self-renew to form identical daughter cells by cell division and differentiate into various types of progenies [19,70]. Over the past twenty years, lung CSCs have been extensively studied and appear as highly heterogeneous populations characterized by multiple markers, such as CD133, CD166, integrin β3, integrin β4, Sca1, CD44 and BMI1 [71,72,73,74]. However, cell of origin is not necessarily linked to CSC [18] and whether CSCs arise from transformed progenitors and/or from reprogrammed tumor cells remains elusive [18,68,75].

Few elements argue that cell of origin might be involved in CSCs. First, longitudinal comprehensive analysis of LUAD has uncovered the rapid emergence of a highly plastic cancer cell state driving LUAD heterogeneity during tumorigenesis [76]. This subpopulation persists in advanced tumors and can functionally give rise to the entire diversity of cell states in the tumor. These results suggest that LUAD does not follow a longitudinal tumor progression program where cancer cells gradually loose lineage characteristics and acquire dedifferentiated features but rather depend on a CSC subpopulation.

Second, cell of origin and CSCs share common markers. BMI1, a member of the Polycomb Repressive Complex 1 (PRC1), is required for BASC self-renewal [30,77,78,79]. Loss of *Bmi1* impaired BASC-mediated bronchiolar repair after Club cell depletion [77]. Aberrant BMI1 expression is a common feature of CSCs [80] and the inhibition of BMI1 abrogates *Kras*-mediated tumorigenesis when it occurs at the early stages and drug resistance and metastatic process when it occurs at the late stages [77,81]. Integrin β4^+^ correlates with tumor aggressiveness [82,83] and defines a tumor cell subpopulation presenting CSC capacities, such as self-renewal, therapy resistance and tumor propagation [72]. As multiple lung stem/progenitor cells express integrin β4, one could presume that integrin β4+ CSCs might arise from progenitors. However, further research will be needed to investigate the consequences of oncogenic expression in different β4^+^ progenitor cells and the subsequent tumor phenotype.

Third, CSCs characteristics are influenced by tumor genetic alterations. Indeed, CSCs markers can differ greatly depending on tumor genotype. Integrin β4^+^ CSCs are enriched in *Kras*-driven tumors harboring the loss of *Tp53* compared to their wild-type counterparts [72]. Likewise, Sca1^+^ cell populations are enriched in CSCs capabilities only in *Kras*/*Tp53* co-mutant-driven tumors. In contrast, the Sca1^−^ cell fraction is enriched in CSCs in *Egfr*-driven tumors [73]. These results demonstrate the importance of the oncogenic driver in determining the CSC phenotype and mirror cell of origin multiplicity. With regard to CSC heterogeneity, one possibility is that cell of origin-derived CSCs as well as cancer cell dedifferentiation-derived CSCs co-exist within the tumor during cancer progression. From a more global perspective, assessing if and how the tumor cell of origin influences CSC molecular features would lead to new avenues in precision medicine.

## 6. Cell of Origin of LUAD: The Interplay with the Immune Microenvironment

Increasing evidence has highlighted the key contribution of the microenvironment in the initiation and progression of lung cancer, which creates a nurturing niche for cancer cells [84]. Comprehensive approaches revealed that the LUAD immune landscape is highly heterogeneous between and within tumors, and exerts a strong selection pressure in early-stage LUAD, producing multiple routes to immune evasion [85]. Lung cancer cells escape from immunosurveillance by reducing antigen presentation, secreting immune-inhibitory cytokines, upregulating immune checkpoints and stimulating immunosuppressive cell subsets [86]. The analysis of the repertoire of immune infiltrating cells has shown that compared to normal adjacent tissue, NSCLC elicits a robust immune response with an increase in T cell infiltration, particularly regulatory T cells (Tregs) [87]. Tregs play a critical role in immune tolerance by inhibiting CD4+ and CD8+ T cell functions [87]. The comparison of immune cells infiltrating tumors between LUAD and LUSC revealed distinct immune phenotype. Neutrophils are more prevalent in LUSC whereas LUAD tumors harbor more macrophages [88].

Emerging evidence suggests that oncogenic drivers influence tumor immune infiltration [89]. *Kras*-driven LUAD are typically characterized by an immunosuppressive state. Indeed, oncogenic *Kras* shapes the immune microenvironment by modulating chemokines, cytokines and growth factors expression with subsequent recruitment of neutrophils and myeloid-derived suppressor cells (MDSCs), creating an immunosuppressive tumor microenvironment [89]. In addition, *Kras* co-occurring mutations are critical factors dictating distinct immune phenotypes and therapeutic vulnerabilities within *Kras*-mutant driven LUAD [90]. The co-activation of the two oncogenes *Kras* and *Myc* cooperates to drive tumorigenesis by eliciting a stromal reprogramming through the activation of epithelial-derived signaling molecules CCL9 and IL-23, which mediate macrophage recruitment and T, B and NK cells exclusion [91]. The immune infiltration comparison of tumors harboring *Kras/Tp53* and *Kras/Stk11* co-mutations uncovered distinct immune phenotypes. TP53 loss of function tumors present the extensive infiltration of cytotoxic CD8+ T cells, as well as the high expression of interferon (IFN)-dependent genes and IFN-induced PD-L1 [52,62,90,92]. In contrast, the inactivation of *Stk11* is associated with an immunosuppressive microenvironment and leads to the accumulation of neutrophils with T cell-suppressive effects, an increase in T-cell exhaustion markers and the secretion of tumor-promoting cytokines [90,92,93,94]. As *Stk11* and *Tp53* co-mutations interplay with the KRAS-driven tumor cell of origin, these results suggest that the tumor cell of origin may influence the tumor immune landscape. In this regard, recent reports compared cell-type specific CC10 and SPC-derived *Kras/Tp53*, *Kras/Stk11* and *Kras/Keap1* co-mutant tumors. They demonstrated that alveolar SPC-derived tumors are highly infiltrated with CD45^+^ cells and depend on tumorigenic macrophages to survive compared to the bronchiolar CC10-derived tumors, regardless of their genetic alterations [52,53,90]. Thus, the tumor immune microenvironment is directly influenced by the tumor cell of origin and not the genetic alterations themselves (Figure 3).

The heterogeneity of immune infiltration in LUAD has significant clinical implications. Recently, cancer immunotherapy has transformed the therapeutic landscape of LUAD. Immunotherapies targeted against the programmed death ligand 1 (PD-L1) and its receptor (PD-1) have dramatically improved the survival of a subpopulation of patients [95]. Retrospective analysis uncovered that immune checkpoint inhibitors (ICIs) responsiveness is influenced by tumor genetic alterations. While *EGFR* mutation leads to the inefficacy of ICIs, KRAS mutant tumors are the best responders [56,96]. However, the underlying association between *KRAS* mutation and immune responses remains unclear. *KRAS* co-occurring mutations are critical factors dictating ICIs tumor response [90,97]. In particular, co-mutation in *TP53* generally upregulates PD-L1, while tumors harboring *STK11* co-mutation are frequently negative for PD-L1 expression [90]. STK11 loss is involved in the suppression of interferon genes (STING), determining a decreased expression of type I interferon genes and chemokines that facilitate T-cell recruitment [98]. Consistent with these results, LKB1 and p53 are predictive biomarkers for ICIs responsiveness [90]. However, not all patients with *KRAS/TP53* mutant tumors respond to ICIs. Thus, response prediction is crucial to stratify tumors and guide precision medicine. While several innate and adaptive ICIs resistance mechanisms have been identified, including the upregulation of alternative immune checkpoints (TIGIT, TIMP-3 and LAG3 [86,99]) and neutrophil infiltration^2^, whether tumor cell of origin could predict ICI resistance has not been intensively investigated yet. However, as the cell of origin influences immune infiltration, and as bronchiolar tumors are infiltrated with T cell suppressive neutrophils, one could assume that the tumor cell of origin could predict ICIs responsiveness and orient precision medicine.

## 7. Future Directions/Conclusions

LUAD tumor heterogeneity mirrors the myriad of factors that are involved in tumor initiation and progression. Not only do LUADs arise from multiple cells-of-origin, but many oncogenic drivers and co-occurring mutations, as well as the dysregulation of the microenvironment, influence tumor behavior. While emerging new technologies provide a more precise picture of lung cancer initiation and evolution, multiple questions still remain. As the nature and the intensity of the injury determine progenitor response, whether this holds true for tumorigenesis is still unknown. Thus, the specific impact of chronic smoke exposure, airborne pollutants and lung microbiota must be investigated with regards of the tumor-cell of origin. The emergence of the complex lung organoid will allow to precisely decipher the involvement of these different cancer triggers on progenitor transformation. How the tumor cell of origin interplays with CSCs and influences the microenvironment are also important questions to answer to better apprehend lung cancer progression and therapy resistance.

From a more global perspective, a better understanding of the tumor cell of origin will provide clues for early detection, thereby improving patient survival and identifying novel pathways allowing for precision medicine.

## Figures and Tables

**Figure 1 cancers-14-01759-f001:**
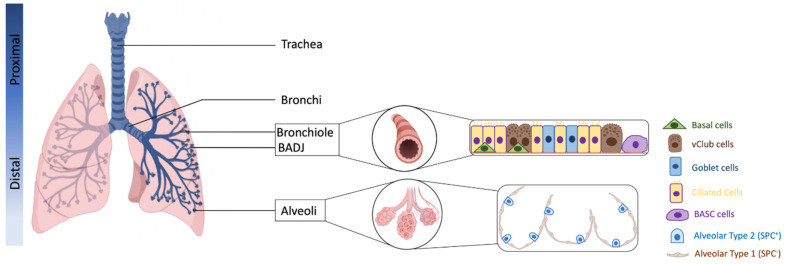
Schematic anatomy of the lung and cellular composition of the airway epithelium (created with BioRender.com, accessed on 15 February 2022).

**Figure 2 cancers-14-01759-f002:**
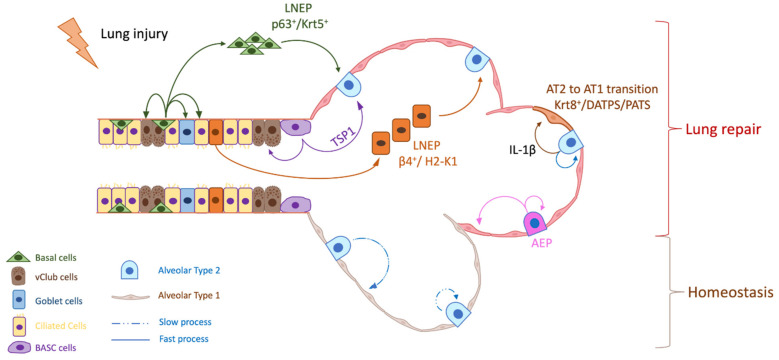
Multiple progenitors are involved in the lung epithelium repair mechanism (Created with BioRender.com, accessed on 15 February 2022). AEP: Alveolar Epithelial Progenitor, AT1: Alveolar Type 1, AT2: Alveolar Type 2, β4: Integrin Beta 4, DATPS: Damage-Associated Transient Progenitors, IL-1β: Interleukin 1 Beta, Krt5: Keratin 5, Krt8: Keratin 8, LNEP: Lineage-Negative Epithelial stem/Progenitor, PATS: Pre-Alveolar type-1 Transitional cell state, TSP1: Thrombospondin-1.

**Figure 3 cancers-14-01759-f003:**
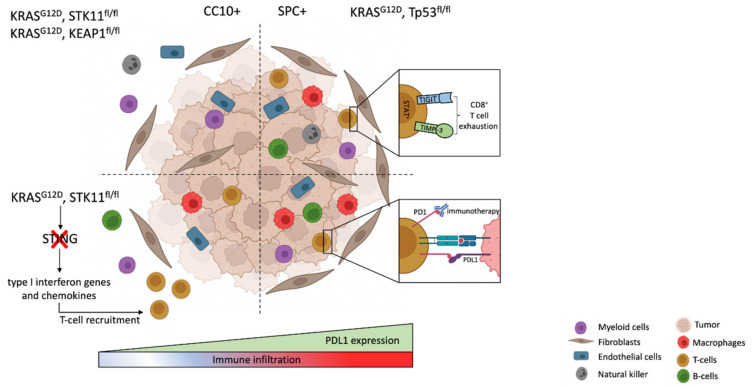
Cell of origin determines the immune microenvironment (created with BioRender.com, accessed on 15 February 2022).

**Table 1 cancers-14-01759-t001:** Summary of LUAD mouse model to distinct cells of origin.

Genetics	Inducer	Target Cells	Inflammation	Tumor Type/Location	Reference
*LSL-Kras^G12D^*	Ad5–Cre	lung epithelial cells	Yes	LUAD/papillary	Kim et al. [30]
*LSL–Kras^G12D^; Trp53^Flox/+^; SPC–CreER; Rosa26R–fGFP*	Tam	SPC+ cells (BASC and AT2)	No	LUAD in alveolar space	Xu et al. [47]
*LSL–Kras^G12D^; Trp53^Flox/+^; CC10–CreER; Rosa26R–fGFP*	Tam	CC10+ cells (BASC and Club)	No	Hyperplasia in the BADJ	Xu et al. [47]
*LSL–Kras^G12Vg^*	Tam	lung epithelial cells	No	LUAD in alveolar space	Mainardi et al. [48]
*LSL–Kras^G12Vg^; Sca1–Cre*	Sca1 expression	Sca1+ cells	no	CC10+ tumors in the bronchiole/BADJ/aleoli	Mainardi et al. [48]
*GFP; Sftpc–CRE*	urethane	lung epithelial cells	yes	LUAD in alveolar space and in the airway	Spella et al. [49]
*GFP; Ccsp–CRE*,	urethane	lung epithelial cells	yes	LUAD in alveolar space and in the airway	Spella et al. [49]
*LSL–Kras^G12D^; Trp53^F/F^*	Ad5–SPC–Cre	SPC+ cells (BASC and AT2)	yes	LUAD in alveolar space	Sutherland et al. [50]
*LSL–Kras^G12D^; Trp53^F/F^*	Ad5–CC10–Cre	CC10+ cells (BASC and Club)	yes	LUAD in alveolar space and BADJ	Sutherland et al. [50]
*LSL–Kras^G12D^; Trp53^F/F^*	Ad5–Cre	lung epithelial cells	yes	LUAD in alveolar space and BADJ	Yin et al. [51]
*LSL–Kras^G12D^; Trp53^F/F^; PRKCi^F/F^*	Ad5–Cre	lung epithelial cells	yes	LUAD in alveolar space	Yin et al. [51]
*LSL–Kras^G12D^; STK11^F/F^*	Ad5–SPC–Cre	SPC+ cells (BASC and AT2)	yes	LUAD in alveolar space	Nagaraj et al. [52]
*LSL–Kras^G12D^; STK11^F/F^*	Ad5–CC10–Cre	CC10+ cells (BASC and Club)	yes	lung adenosquamous and LUSC	Nagaraj et al. [52]
*Kras^G12D^; KEAP1^F/F^*	Ad5–SPC–Cre	SPC+ cells (BASC and AT2)	yes	LUAD in alveolar space	Best et al. [53]
*Kras^G12D^; KEAP1^F/F^*	Ad5–CC10–Cre	CC10+ cells (BASC and Club)	yes	LUSC in the airway	Best et al. [53]

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
