# Peer review of "Lung Adenocarcinoma Tumor Origin: A Guide for Personalized Medicine"

_cancers, 2022, doi:10.3390/cancers14071759_

Round 1

Reviewer 1 Report

Manuscript ID: cancers-1630009

Type of manuscript: Review

Title: Lung tumor origin, a guide for personalized medicine

Authors: Laetitia Seguin *, manon durandy, CHLOE C FERAL *

Submitted to section: Molecular Cancer Biology,

https://www.mdpi.com/journal/cancers/sections/Molecular_Cancer_Biology

Understanding New Therapeutic Options and Promising Predictive Biomarkers for

Lung Cancer Patients. A Selection of Papers from the Third Joint Meeting on

Lung Cancer of the FHU OncoAge (Nice, France) and the MD Anderson Cancer

Center (Houston, TX, USA)

https://www.mdpi.com/journal/cancers/special_issues/Understanding_Therapeutic_Biomarkers_Lung

In this manuscript, the authors reviewed the current knowledge about the cell of origin of lung adenocarcinoma (LUAD). They thoroughly describe several lung progenitor cells as candidates to give rise to LUAD which are affected by several factors including mutational status and immune tumor microenvironment, thus leading to the high heterogeneity of this fatal disease. It is highlighted that further investigation on the cell of origin and their impact on cancer stem cells will lead to a better understanding of LUAD improving personalized medicine and ultimately enhancing patient survival.

General comments:

Seguin et al. present a very comprehensive review on the cell of origin of LUAD covering many aspects leading to tumor heterogeneity. The cited literature seems accurate and recent. The summaries at the end of each section make the manuscript comprehensible. The text is well structured and provides such detail that readers who do not come from the lung field, can easily understand the content as well.

Since the review mainly discusses LUAD, one histologic subtype of lung cancer, this should be obvious in the manuscript title. Speaking of lung tumor origin seems too general.

The authors should highlight throughout the text what studies were done on mice and which findings were retrieved on human tissues (especially in sections 1.-3.). It’s not always obvious, although it is of importance to know the difference. Perhaps correct case sensitivity for human and murine gene names would help. Please also revise the manuscript for italic spelling of gene names.

For completeness, also those cells capable of repopulating the airways including basal cells and secretory cells should be thoroughly discussed. Section 2 focusses on differentiation to alveolar cells only.

In section 3., there is a paragraph repeated 4 times which includes experimental findings (not appropriate for a review). The authors should please correct this.

The displayed figures nicely illustrate the text. However, due to its numerous abbreviations Figure 2 would benefit from a caption in addition to the figure title. The table neatly summarizes the studies conducted on various mouse models investigating the LUAD cell of origin.

For consistency, the authors should revise the references including missing author names, years, and capitalization, e.g., in references 5, 30, 33, 35, 44.

Taken together, Seguin et al. provide a decently written manuscript with only a few flaws that should be revised. The authors arise important questions for further investigation in LUAD and conclude the manuscript with vital research questions for precision medicine.

Specific comments:

  • Lines 43-46: please include the reference
  • Lines 78-79: please include the reference
  • Lines 292-293: please include the reference
  • Line 420: not quite clear, it should probably read something like “to normal adjacent tissue”

Author Response

REVIEWER 1

General comments:

Seguin et al. present a very comprehensive review on the cell of origin of LUAD covering many aspects leading to tumor heterogeneity. The cited literature seems accurate and recent. The summaries at the end of each section make the manuscript comprehensible. The text is well structured and provides such detail that readers who do not come from the lung field, can easily understand the content as well.

We thank the reviewer for his/her feedback. We answered point-by-point below (in blue) inserted in the review.

Since the review mainly discusses LUAD, one histologic subtype of lung cancer, this should be obvious in the manuscript title. Speaking of lung tumor origin seems too general.

According to the reviewer 1 comment, the new title is : Lung adenocarcinoma tumor origin, a guide for personalized medicine

The authors should highlight throughout the text what studies were done on mice and which findings were retrieved on human tissues (especially in sections 1.-3.). It’s not always obvious, although it is of importance to know the difference. Perhaps correct case sensitivity for human and murine gene names would help. Please also revise the manuscript for italic spelling of gene names.

According to the reviewer comment, we highlighted throughout the text when studies are done on mice or human tissues. We also modified human and murine gene names accordingly.

For completeness, also those cells capable of repopulating the airways including basal cells and secretory cells should be thoroughly discussed. Section 2 focusses on differentiation to alveolar cells only.

We thank reviewer 1 for this comment. We fully agree that basal cells and secretory cells are crucial in repopulating the airways. Nonetheless, in the scope of this review, we made the choice of focusing only on alveolar cells as i) excellent previous reviews are already published on the subject (ex. Basal-like Progenitor Cells: A Review of Dysplastic Alveolar Regeneration and Remodeling in Lung Repair stem cell report, de Mello Costa, Weiner, and Vaughan) and ii) since the review focuses on LUAD, we concentrated on their cell of origin (alveolar cells).

In section 3., there is a paragraph repeated 4 times which includes experimental findings (not appropriate for a review). The authors should please correct this.

We corrected the mistake

The displayed figures nicely illustrate the text. However, due to its numerous abbreviations Figure 2 would benefit from a caption in addition to the figure title. The table neatly summarizes the studies conducted on various mouse models investigating the LUAD cell of origin.

We added a caption for Figure 2

For consistency, the authors should revise the references including missing author names, years, and capitalization, e.g., in references 5, 30, 33, 35, 44.

We corrected the references

Taken together, Seguin et al. provide a decently written manuscript with only a few flaws that should be revised. The authors arise important questions for further investigation in LUAD and conclude the manuscript with vital research questions for precision medicine.

Specific comments:

  • Lines 43-46: please include the reference
  • Lines 78-79: please include the reference
  • Lines 292-293: please include the reference
  • Line 420: not quite clear, it should probably read something like “to normal adjacent tissue”

Done for all (in this new version, line numbers have change, i.e line 420 is now line 517)

Reviewer 2 Report

Seguin et al. described the cell of origin of lung adenocarcinoma in their review article. The contents are good and figures and a table are adequate, therefore I recommend to publish this article.  The followings are minor points.

  1. The 1st paragraph of 3rd section (Cell-of-Origin of LUAD, Multiple Possibilities) is repeated in the 2nd, 3rd, and 4th paragraphs of the section. Please revise the repetition.  Moreover, “lveolar pithelial rogenitor (AEP)” should be revised to “alveolar epithelial progenitor (AEP)”.
  2. In the Table 1, in the column of Reference, not only author name but also reference number should be added.

Author Response

Please see the attachment (part reviewer2)
